# Understanding How Genetic Mutations Collaborate with Genomic Instability in Cancer

**DOI:** 10.3390/cells10020342

**Published:** 2021-02-06

**Authors:** Laura J. Jilderda, Lin Zhou, Floris Foijer

**Affiliations:** European Research Institute for the Biology of Ageing (ERIBA), University of Groningen, University Medical Centre Groningen, 9713 AV Groningen, The Netherlands; l.j.jilderda@umcg.nl (L.J.J.); l.zhou@umcg.nl (L.Z.)

**Keywords:** aneuploidy, chromosomal instability, genome wide screens, cancer

## Abstract

Chromosomal instability is the process of mis-segregation for ongoing chromosomes, which leads to cells with an abnormal number of chromosomes, also known as an aneuploid state. Induced aneuploidy is detrimental during development and in primary cells but aneuploidy is also a hallmark of cancer cells. It is therefore believed that premalignant cells need to overcome aneuploidy-imposed stresses to become tumorigenic. Over the past decade, some aneuploidy-tolerating pathways have been identified through small-scale screens, which suggest that aneuploidy tolerance pathways can potentially be therapeutically exploited. However, to better understand the processes that lead to aneuploidy tolerance in cancer cells, large-scale and unbiased genetic screens are needed, both in euploid and aneuploid cancer models. In this review, we describe some of the currently known aneuploidy-tolerating hits, how large-scale genome-wide screens can broaden our knowledge on aneuploidy specific cancer driver genes, and how we can exploit the outcomes of these screens to improve future cancer therapy.

## 1. Introduction

During each cell division, a cell’s genome is replicated, after which all chromosomes need to be properly distributed over the two emerging daughter cells. Continuous errors during chromosome segregation, also known as chromosomal instability (CIN), leads to cells with chromosome numbers that deviate from the euploid karyotype, a state defined as aneuploid [1]. Aneuploidy is highly detrimental during development, which is reflected by the fact that it is the leading cause of spontaneous abortion and mental retardation in humans [2]. When induced experimentally, aneuploidy negatively affects cellular fitness by reducing cell growth and inducing metabolic and proteotoxic stress [3,4,5,6]. However, aneuploidy is a hallmark of cancer [7,8], a disease characterized by uncontrolled proliferation. This apparent contraction, also known as the aneuploidy paradox [9], suggests that aneuploid cells must activate ‘aneuploidy-coping’ mechanisms in order to adopt a malignant fate. Therefore, the cellular stresses imposed by aneuploidy are considered to be attractive targets for therapeutic intervention.

The currently-known aneuploidy-tolerating hits and pathways have mostly been identified from small scale screens or through educated guesses using model systems for aneuploid non-transformed cells or cancer cell lines. While these findings are key for our understanding of the biology of aneuploid cells, they unlikely draw the complete picture of aneuploidy tolerance pathways. This is partly because the screens and model systems used are biased towards pathways that we already understand reasonably well. Furthermore, these experiments were mostly done in cultured cells and thus do not account for the in vivo malignant transformation process and interactions between tissues. To acquire a more comprehensive overview of how cells adapt to aneuploidy during malignant transformation, unbiased genome-wide in vivo screens that carefully compare the tumor drivers between aneuploid and euploid cancers are a next important step forward. In this review, we discuss a selection of the model systems for stable aneuploidy and briefly touch upon models for ongoing CIN. We discuss some of the main findings from these models and how they have been used to identify aneuploidy-tolerating pathways. Finally, we describe several types of large-scale mutagenesis screens, what their advantages and limitations are, and how they can be exploited for the unbiased identification of aneuploidy-tolerating mechanisms.

## 2. Model Systems for Stable Aneuploidy

One way to model the consequences of aneuploidy is by inducing chromosomal instability in the target cells, which leads to an aneuploid cell population comprising cells with various karyotypes (Figure 1A). While CIN and aneuploidy have many overlapping characteristics, they are different concepts and CIN cells may have their own targetable vulnerabilities [1]. It is therefore important to distinguish models for stable aneuploidy and models for ongoing CIN. To circumvent the complications that arise from karyotype heterogeneity within the cell population caused by ongoing CIN, several stable aneuploid cell models were engineered from euploid controls (Figure 1B–D) a selection of which are summarized hereunder.

### 2.1. Aneuploid Yeast Models

The budding yeast *Saccharomyces cerevisiae* has been widely used to study aneuploidy and its cellular effects (Figure 1B). To create aneuploid yeast strains in a haploid background (i.e., disomes), a KAR1 deficient yeast strain was crossed with wildtype yeast to prevent nuclear fusion [3]. During such matings, individual chromosomes are occasionally transferred from one nucleus to the other, leading to aneuploid daughters, which were selected for using selectable markers. This method yielded 20 yeast strains bearing one or multiple extra copies for almost all of the yeast chromosomes, which were exploited to assess the consequences of aneuploidy, as is further discussed below [3].

### 2.2. Stable Aneuploid Human Cell Lines

To study stable aneuploidy in human cell lines, mouse cells containing an ectopic human chromosome with a selectable marker were exposed to prolonged colcemid treatment and subsequently released to induce the formation of micronuclei [10,11,12] (Figure 1C). Individual micronuclei were isolated and separated by centrifugation after which they were fused with recipient human cells. Recipient cells with the extra human chromosome were selected for using a selection cassette. This technique yielded various stable aneuploid human cell lines including defined trisomic and even tetrasomic RPE1, HE35, and colorectal cancer HCT116 and DLD1 cell lines, which allowed for the studying of effects of aneuploidy using isogenic aneuploid and euploid cell lines [6,13].

### 2.3. Stable Aneuploid Mouse Embryonic Fibroblasts

As an alternative strategy to generating stable aneuploid mammalian cell lines, the Amon lab generated primary mouse embryonic fibroblasts (MEFs) that carry one additional chromosome (trisomic, Ts) [4] (Figure 1D). These MEF lines were generated by intercrossing mice homozygous for two different Robertsonian translocations. Male offspring that carried both translocations were then mated with wild-type mice resulting in a percentage of the progeny with a trisomy for the chromosome common to the two Robertsonian translocations. All trisomic embryo’s (except for Ts19 mice, which survived a few days postnatally) died in utero as expected, but many embryos developed past embryonic day 12.5 allowing for MEF isolation. These efforts yielded MEFs trisomic for chromosome 1, 13, 16, and 19.

## 3. Models for Ongoing Chromosomal Instability

Most studies that identified aneuploidy-tolerating mechanisms made use of stable aneuploid cell lines. Therefore, in these lines, the aneuploidy response is expected to manifest homogeneously throughout the population. However, the karyotypes of aneuploid cancer cells are often much more heterogeneous, a result of ongoing CIN. Therefore, to understand how the aneuploidy paradox plays out in vivo, ensuing cells with a CIN phenotype probably is a more physiologically relevant approach. One way to induce ongoing CIN is by alleviating the spindle assembly checkpoint (SAC). The SAC acts as a safeguard mechanism in mitosis that coordinates chromosome attachment to the mitotic spindle. To prevent CIN, the SAC halts cells in metaphase to prevent sister chromatid separation when chromosome pairs are unattached or improperly attached [14,15]. Conversely, to provoke a CIN phenotype, the SAC can be alleviated by the use of drugs that interfere with SAC proteins such as the MPS1 inhibitors AZ3146 [16] and Reversine [17], or the Mad2 inhibitor M2I-1 [18] by genetically alleviating SAC proteins such as Mps1, Mad2, or Bub1 [19,20,21]. Alternatively, CIN can for instance be induced by amplification of centrosomes [22,23,24], by interfering with kinetochore structure [25], by disrupting the cohesin complex [26], or by deregulating microtubule polymerization rates [27].

When comparing the findings between models for stable aneuploidy and ongoing CIN (the latter extensively discussed elsewhere [28,29]), it appears that similar pathways are deregulated. However, mapping the consequences for ongoing CIN is complicated by the intrinsically heterogenous nature of the models. This heterogeneity complicates separating the stresses triggered by the ongoing CIN and those triggered by the random aneuploidies provoked by CIN. One solution to this problem is to engineer models for reversible CIN, for instance, as previously done in Drosophila [30], which allows to uncouple the aneuploidy and CIN response in vivo.

However, in this review, we mainly focus on the stress pathways identified in stable aneuploid cell models, as is further discussed below.

## 4. Early Findings from Model Systems

An important first finding from stable aneuploidy model systems is that aneuploid cells have a significant proliferative disadvantage compared to their euploid counterparts [3,4,6,13]. Furthermore, all aneuploid models showed strong metabolic changes and induction of stress responses, indicating that aneuploidy decreases cellular fitness, and thus is expected to suppress tumorigenesis. However, aneuploidy frequently occurs in human cancer [7,8], a disease characterized by increased cell proliferation. These contradicting observations are frequently referred to as the ‘aneuploidy paradox’ [9] and suggest that aneuploid cells need to overcome certain barriers to become malignant. As overcoming these stresses might be an important step during tumorigenesis, modulators of aneuploidy-imposed stresses are considered attractive targets for therapeutic intervention [31]. Below, we discuss some of these stresses and elaborate on how these findings have broadened our knowledge on how cancer cells deal with aneuploidy.

### 4.1. Gene Expression Changes

Studies in various aneuploid yeast strains revealed that aneuploidy leads to deregulation of the transcriptome [3], yielding a gene expression signature that is characteristic of the yeast environmental stress response (ESR). This ESR signature is characterized by an increased expression of ribosomal biogenesis genes and genes involved in nucleic acid metabolism. Conversely, genes involved in carbohydrate metabolism displayed decreased expression. A similar but not identical response was found in human cells when comparing DNA, mRNA, and protein levels of euploid and aneuploid human cell lines [6]. For this purpose, multiple human tri-and tetrasomic cell lines were transcriptionally profiled and compared to aneuploid human cancer cell lines. This led to the identification of a general transcriptional aneuploid response pattern (ARP) [13]. This ARP signature includes the upregulation of genes involved in the endoplasmatic reticulum (ER) and Golgi-related pathways, lysosome/lytic vacuoles, MHC protein complex, antigen processing, and a downregulation of genes involved in DNA and RNA metabolism and ribosome-related pathways. The ARP thus might point towards altered growth requirements of aneuploid cells, which could represent a potential therapeutic target in aneuploid cancer treatment.

While this general ARP pattern seems to hold true for several aneuploid cell strains [13] and is also observed in vivo (e.g., in aneuploid basal epidermal cells) [32], part of this aneuploidy signature is reverted in aneuploid cancers. For instance, while cultured aneuploid cells appear to have decreased expression of ribosomal genes [13,32], expression these genes is upregulated in a model for CIN-driven T-ALL [33]. Interestingly, as discussed above, stable aneuploid yeast strains also exhibit increased expression of ribosome genes [3], suggesting that yeast cell biology represents the biology of an in vivo cancer cell more than the biology of mammalian cell lines, at least in its response to aneuploidy.

While aneuploidy clearly triggers expression changes and while there is a remarkable correlation between gene/chromosome copy number changes and gene expression in stable aneuploid cells [34], as well as in aneuploid cancers [33,35,36], there is also evidence that the expression of a small fraction of aneuploid genes is somehow dose-compensated [36,37,38]. This suggests that some genes need to be epigenetically modulated upon aneuploidization, potentially uncovering a targetable vulnerability of aneuploid cancer cells.

### 4.2. Proteotoxicity

As aneuploidy leads to increased gene expression and thus increased protein production from the aneuploid chromosomes (in case of chromosome gains) [39], aneuploidy disrupts the protein complex stoichiometry [6,40]. The resulting excess of ‘aneuploid’ proteins causes proteotoxic stress, such as impaired protein folding [41], unscheduled protein degradation [42,43] and increased protein aggregation [44]. Together, these processes are believed to reduce cellular fitness of aneuploid cells [45].

Conversely, reducing proteotoxic stress levels was found to favor the survival of aneuploid cells. For instance, fast growing aneuploid yeast strains often harbor inactivating mutations in *UBP6* [46]. *UBP6* deubiquitinates substrates at the proteasome, which leads to recycling of ubiquitin and rescue of proteasome substrates from degradation. Thus, the absence of *UBP6* leads to an increased clearance of proteins, which is beneficial to aneuploid cells. Together, these findings reveal that proteotoxic stress is an adverse effect of aneuploidy and, moreover, that *UBP6* mutations allow aneuploidy tolerance in yeast by increasing proteasomal degradation. Conversely, the deletion of *UBP3*, encoding the deubiquitinase *UBP3*, exerts a significant negative fitness penalty to aneuploid yeast cells [47]. As *UBP3* is required for efficient functioning of the ubiquitin-proteasome system, its deletion impairs protein degradation, thereby further increasing proteotoxicity in aneuploid yeast. This aneuploidy-related liability is conserved in human cells, as depletion of the human homolog of *UBP3* and *USP10* also reduces the fitness of chromosomal instable RPE1 cells [48].

Another cause for the observed proteotoxicity in aneuploid cells is a decrease in heat shock protein (HSP) (i.e., 90-mediated protein folding), which ultimately leads to protein aggregation in the cytoplasm [41,44]. Aneuploid yeast and human cells are therefore more sensitive to HSP90 inhibiters such as 17-AAG [3,41,42]. Conversely, increasing heat shock protein 1 (HSF1) levels rescues the effects of aneuploidy on HSP90 expression and resulting proteotoxicity in human cells, thereby identifying HSF1 as an aneuploidy tolerating gene [41].

Finally, autophagy, a process that removes surplus and damaged proteins and organelles, is upregulated in aneuploid cells [6]. Stable aneuploid human colon cancer cells display increased numbers of LC3 foci, an autophagy marker [43], and upregulated p62-dependent autophagy. Lysosomes play an essential role in autophagy, and protein aggregates accumulate in the lysosomes of chromosomal instable cells. This triggers a lysosomal stress response that can be relieved by activating the transcription factor TFEB, which increases expression of autophagy genes [49]. Thus, to cope with aneuploidy-imposed proteotoxicity, aneuploid cells require an upregulation of autophagy, which potentially can be exploited in the treatment of aneuploid cancers through inhibitors of autophagy such as chloroquine [42] and bafilomycin A [13,50].

Taken together, aneuploid cells heavily rely on their protein quality control machinery, which renders them more sensitive to compounds interfering in these processes. It is therefore of the utmost importance to further investigate these potential targetable vulnerabilities of aneuploid cancer cells.

### 4.3. Metabolic Stress

Besides proteotoxic stress, aneuploid cells also suffer from metabolic stress [3,4,42]. This is exemplified by the fact that aneuploid cells are much more sensitive to the energy stress-inducing compound AICAR than their euploid counterparts [42]. Indeed, AICAR exacerbates aneuploidy-imposed energy stress by activating AMP-activated protein kinase, which results in the efficient killing of stable trisomic MEFs. Conversely, euploid MEFs continue to proliferate when exposed to the same concentrations of AICAR.

Similar to AICAR, the glucosylceramide synthase inhibitor, DL-PDMP, also selectively inhibits the proliferation of Ts13 MEFs [51]. DL-PDMP is a ceramide analogue that inhibits glucosylceramide synthase, thereby decreasing ceramide glycosylation. This causes an accumulation of ceramides, which inhibits proliferation and promotes apoptosis of aneuploid MEFs in highly aneuploid colorectal cancer cells rather than their euploid counterparts. Aneuploid cells possibly express higher ceramide levels than euploid control cells, thereby sensitizing them to compounds that further increase these ceramides. However, inhibition of sphingolipid synthesis was also shown to impair the fitness of aneuploid yeast [52], suggesting that aneuploid cells critically rely on perfect titration of this pathway for their survival.

### 4.4. Inflammatory Response

Finally, while the above-described mechanisms mostly impinge on cell-intrinsic mechanisms, accumulating evidence reveals an important role for the immune system in controlling the propagation of aneuploid cells. For instance, tri- and tetrasomic RPE1, HCT116, and DLD1 cell lines display upregulated interferon (IFN) signaling and increased expression of proteins involved in MHC protein complex and antigen processing [13,53]. Similarly, aneuploid RPE1 cells exhibit an upregulation of pro-inflammatory cytokines such as IL-6, IL-8, and CCL2, which could promote immunosurveillance [54].

DNA is normally compartmentalized to the nucleus, but following chromosome mis-segregation, chromosomes end up outside of the main nucleus in a smaller so-called micronuclei [55]. When these micronuclei rupture, genomic DNA is released in the cytosol, which activates the cytosolic nucleic acid sensor cyclic guanosine monophosphate (GMP)-adenosine mononphosphate (AMP) synthase (cGAS) [55,56]. Activated cGAS then stimulates type I interferon signaling through a stimulator of interferon genes (STING) and the downstream transcription factor IRF3 ultimately leading to an inflammatory response of the micro-nucleated cell [57]. The cGAS-STING thus plays a pivotal role in anti-tumor immunity and STING signaling indeed appears to be altered in a variety of cancers [58,59]. Paradoxically, active STING was found to be crucial for the metastasis of tumors exhibiting a CIN phenotype. It therefore requires further work to fully understand in which context cGAS-STING signaling is tumor promoting or suppressive [60].

Most described models that study aneuploidy have looked into what allows aneuploidy tolerance, focusing into what cellular changes or stresses are induced in cells by aneuploidy. However, the majority of these efforts were educated guesses or small-scale screens, particularly the studies in mammalian cells. A next important step forward is therefore to search for the processes that transform aneuploid cells into aneuploid cancer cells in a completely unbiased fashion, for instance through functional genetic in vivo screens that systematically compare the drivers between aneuploid and euploid cancers.

Below, we describe which types of genetic screens can be used for cancer gene identification and what their main advantages and disadvantages are (Table 1).

## 5. Genetic Screens to Identify Cancer-Collaborating Hits

### 5.1. Chemical Mutagenesis

One way to search for cancer genes is by randomly mutagenizing the genome of cells and next search for cancer(-like) phenotypes. The alkylating chemical N-ethyl-N-nitrosourea (ENU) is a potent agent to induce mutations in mice [61], and has been widely used to find drivers for human disorders in mouse models [62]. For ENU mutagenesis screens, male mice are treated with ENU and crossed with wild type females to produce offspring that can be assayed for dominant mutations. Recessive mutations can be identified by intercrossing or backcrossing pedigree [62]. All offspring are examined for the relevant disease phenotype by studying their behavior, physiology, or dysmorphology of the mice, after which the gene responsible for the observed phenotype in mice of interest can be identified through laborious positional cloning [62].

ENU mutagenesis has been used to find drivers for many human disorders and to better understand gene function and complex biological systems [62]. For instance, it was exploited to find mutations that lead to resistance to the EGFR monoclonal antibody Cetuximab in colorectal cancer patients [63] and provided new leads to treat Cetuximab resistant tumors. In another example, ENU mutagenesis was used to identify double strand break (DSB) repair genes in mice when screening for in vivo chromosome damage using micronuclei as a readout [64]. This screen identified a recessive mutation that leads to elevated levels of spontaneous and radiation- or mitomycin C-induced micronuclei.

As exemplified above, ENU mutagenesis can be applied to map pathways in biological systems as well as to better understand cancer susceptibility. However, a major disadvantage is the large cohort of mice required for such screens and the laborious cloning needed to identify the causative genes. When retroviral insertional mutagenesis screens were introduced, these rapidly became an attractive alternative to ENU mutagenesis screens, as mapping viral integration sites is much less laborious than positional cloning of ENU-mediated mutations in the genomic DNA [65,66].

### 5.2. Retroviral Insertional Mutagenesis

Retroviruses consist of an RNA genome that replicates via a provirus (DNA intermediate). The provirus randomly integrates into the host genome, which results in mutagenesis at the integration site [67]. Proviral integrations can lead to either activation of a gene, posttranscriptional dysregulation, gene inactivation, or gene truncation when integrating near to or in a gene [68]. This gene-disrupting feature was exploited for screening purposes as the resulting deregulation of affected chromatin could lead to a growth advantage by either activation of an oncogene or inactivation of a tumor-suppressor gene so that cells carrying that integration would be enriched in the tumor cell population [69]. Viral integration sites can easily be identified by cloning the provirus and adjacent cellular DNA from the tumor DNA [67]. Retroviral insertional mutagenesis has also been extensively used to identify cancer genes in mouse and human model systems.

Various retroviruses have been used for this purpose. For instance, the Moloney murine leukemia virus (MoMuLV) was used to identify collaborating drivers in BXH2- and AKXD-predisposed mice leading to BXH2 myeloid leukemia and AKXD lymphomas [70]. Subsequent mapping of hundreds of proviral insertions revealed several previously-described collaborating drivers as common insertion sites (CIS) in these lymphomas and myeloid leukemia’s, providing proof-of-principle, but, importantly, also new cancer driving genes [70]. Another MoMuLV screen to identify genes that can substitute for *Pim1* and *Pim2* in lymphomagenesis in *Myc* transgenic, *Pim1*, and *Pim2* doubly deficient mice [71], uncovered several CIS (of which 10 belonged to the *Pim* complementation group), which furthered our understanding of *Pim1/2*-driven lymphomagenesis. Similarly, a MoMuLV screen performed in *Cdkn2a*^-/-^ mice identified genes that collaborate with the tumor suppressors *p16^INK4a^* and *p19^ARF^* encoded by the *Cdkn2a* locus [72]. Moreover, the CIS included established cancer drivers and new ones with an enrichment for genes involved in MAPK signaling. Also other tumor viruses can be exploited for insertional mutagenesis screens, such as the mouse mammary tumor virus (MMTV), which led to the identification of various mammary carcinoma genes [73].

In summary, retroviral insertional mutagenesis screens, also known as retroviral tagging screens, can be used to identify genes driving cancer. However, a major disadvantage is that these viruses display a selective cellular tropism, indicating that these viruses can only infect certain tissue types [69]. Therefore, retroviral tagging screens were largely replaced by transposon screens for which it was much easier to target different cell lineages in vivo. This expedited the development many more specific cancer gene screening models.

### 5.3. Transposon Mutagenesis Screens

Transposons are mobile genetic elements that can be used to perform unbiased genetic screens to identify driver genes in human cancers. The most commonly used transposon types in in vivo genetic screens are Sleeping Beauty (SB) [74,75] and Piggyback (PB) transposons [76,77,78]. A major advantage of PB compared to SB is that PB transposons do not leave undesired footprint mutations behind after transposition, which significantly simplifies the identification of the driver genes [79]. Both SB and PB make use of two components: (1) a mutagenic gene trap that can either activate or disrupt gene expression depending on where the transposon integrates [80,81] and (2) the enzyme transposase. The transposase can be ubiquitously or tissue-specifically expressed, the latter permitting tissue-specific mutagenesis. Similar to retroviral tagging, the genomic integration sites are relatively easy to map, for instance using splinkerette-PCR followed by Sanger or next-generation sequencing.

The first transposon screens used ubiquitously expressed SB transposase and identified genes that promote sarcomas and hematopoietic malignancies [80,81]. In later screens, Cre-inducible SB or PB transposase was used to restrict transposon mutagenesis to specific tissues such as B-cells, liver, and the gastrointestinal tract [82,83,84]. In addition, transposon screens were employed to find drivers of many other tumor types including breast, lung, prostate, thyroid, melanoma, and medulloblastoma (reviewed in [85,86,87]). Together, these studies showed that transposon mutagenesis can drive tumorigenesis in wild type or genetically-predisposed backgrounds. Examples include an SB screen performed in *p53*-proficient and *p53-*deficient mice, which identified driver genes of osteosarcoma development and metastasis [88]. Another SB screen performed in mice that carried various colon cancer predisposing mutations [89] identified various new driver genes, which are highly relevant for early and late cancer stages. Furthermore, an SB screen in *Pten* mutant mice identified many new driver genes of triple-negative breast cancer [90]. Similarly, PB mutagenesis was also successfully used to identify cancer genes in hematopoietic and solid malignancies [78]. For instance, a PB screen in *p19^ARF^* deficient mice identified genes that cause drug resistance against MDM2-TP53 inhibitors [91].

While we only highlighted a few examples of transposon screens here, it is clear that, collectively, transposon screens have contributed tremendously to our understanding of the process of tumorigenesis in vivo. However, it is important to note that transposon screens randomly mutagenize the genome. There are also more targeted approaches to inactivate genes, such as RNAi and clustered regularly interspaced short palindromic repeats (CRISPR), that can also be exploited for genetic screens, as is further discussed below.

### 5.4. RNA interference Screens

RNA interference (RNAi) is an evolutionarily conserved mechanism among eukaryotes functioning as a defense mechanism against double stranded-RNA (dsRNA) to target cellular and viral mRNAs [92,93]. As RNA interference leads to degradation of the targeted RNA molecule [94,95,96,97], this mechanism rapidly became exploited as a powerful technology to silence gene products in vitro and in vivo. For this purpose, short interfering RNAs (siRNA) are delivered into mammalian cells to induce transient but efficient and specific down-regulation of target mRNAs [98]. Alternatively, to induce sustained gene-silencing, short hairpin RNAs (shRNA) can be delivered through retroviral and lentiviral vectors [99,100].

In addition to targeted downregulation of individual genes, RNAi technology can also be applied in a high throughput manner. For this purpose, genome-wide shRNA-based libraries were developed that allowed high-throughput RNAi screens in mammalian cells, which can be performed as pooled screens or arrayed screens. In the case of arrayed screens, individual wells with cells are transfected or transduced with individual RNAis/shRNAs and screened for the assessed phenotype. In pooled screens, large cell populations are transduced with shRNA libraries (whole genome or targeting specific gene families) at once. Cells with a phenotype of choice are then selected/enriched for after which the shRNAs driving the phenotype are identified by sequencing the integrated shRNAs. Such pooled shRNA screens have been performed both in vitro and in vivo and mostly focused on genes that contribute to tumorigenesis [100].

For example, one shRNA screen investigating chromatin-modifying enzymes in aggressive castration-resistant prostate cancer showed that knockdown of histone H3K9 demethylase, *KDM3B*, has an antiproliferative effect in cell lines [101]. Another RNAi screen revealed that the loss of *FBW7* is synthetically lethal with an inactivated SAC in colorectal cancer cell lines [102]. Importantly, shRNAi screens can also be used in vivo, exemplified by an shRNA screen that identified over 10 candidate tumor suppressor genes in a mouse lymphoma model [103].

While most shRNA screens were conducted in mammalian systems, shRNAi screens are not restricted to mammalian cells. For instance, an RNAi screen in *Drosophila* used to identify genes that are required for the viability of CIN cells, revealed that genes involved in centrosomal and JNK signaling specifically induce apoptosis in cells with a CIN phenotype. In this screen, Mad2 RNAi was used to alleviate the SAC and provoke CIN, which was combined with individual RNAi’s targeting a set of kinase and phosphatase genes [104]. Some of the identified hits were, at the time of the screen, already being pursued as cancer therapy targets, underscoring that such screens can identify drug targets of clinical significance [105]. Another RNAi screen in cultured Drosophila S2 cells revealed that *HSET*, normally a non-essential kinesin motor, is essential for the survival of cells with extra centrosomes, uncovering *HSET* as a potentially very selective anti-cancer target for cancers displaying centrosome amplification [106].

While RNA interference functional screens are still widely-used, they have slowly begun to lose momentum to CRISPR/ CRISPR associated nuclease 9 (Cas9)-powered functional genetic screens, as is further discussed below.

### 5.5. CRISPR-Cas9 Screens

Clustered regularly interspaced short palindromic repeats (CRISPR)/CRISPR associated nuclease 9 (Cas9) is a powerful genome editing tool that revolutionized the genome editing field immediately after its discovery. The CRISPR-Cas system originates from prokaryotic organisms (e.g., bacteria, archaea) in which it provides acquired resistance against foreign genetic elements such as bacteriophages [107]. Cas9 is a large gene encoding for a single-effector protein that can cleave DNA [108]. The Cas9 protein is guided to specific DNA sequences through a duplex of a CRISPR RNA (crRNA) and trans-activating RNA (tracrRNA) [109] and then cleaves the targeted DNA at the so-called protospacer adjacent motif (PAM) [108]. This prokaryotic immune system was adapted to target DNA in mammalian cells by combining the crRNA and tracrRNA into one single guide RNA (sgRNA) construct [110]. This allows for the targeting of Cas9 to any DNA sequence of choice [110], which then leads to a double-stranded break (DSB) at the targeted DNA sequence, opening up a wide range of possibilities in the field of genome editing and gene therapy [111]. Further adaptations to the CRISPR/Cas9 system led to new CRISPR applications. For instance, a nuclease dead mutant of Cas9 (dCas9) that maintains sgRNA directed binding of specific DNA sequences, was fused to a transcriptional repressor domain to yield a Cas9 protein that can inhibit transcription (CRISPR interference or CRISPRi) [112] or activate transcription when Cas9 is fused to a transcriptional activator domain [113] (CRISPR activation or CRISPRa). ‘Conventional’ CRISPR, as well as CRISPRi/CRISPRa technologies, can be exploited for genome-wide or targeted functional genetic screens, for which many guide RNA libraries (synthetic guide RNA and viral vectors) are available in both academic and commercial domains.

Indeed, many libraries have been used to identify tumor suppressor genes and oncogenes [114], as have drug targets and drug resistance mechanisms [115]. For instance, one recent CRISPR screen in triple negative breast cancer cells showed that inactivation of genes involved in the anaphase-promoting complex, such as *ANAPC4*, *ANAPC13,* and *MAD2L1BP*, confers resistance towards Mps1 inhibitors [116]. In another example, a non-metastatic mouse cancer cell line was transduced with a genome-wide guide RNA library encompassing 70,000 sgRNAs. The CRISPR polyclonal cell pool was transplanted into immunocompromised mice and recipient mice were screened for metastatic cancer. Lung metastases and late-stage primary tumors were enriched for sgRNAs targeting a small set of genes that included genes involved in pro-apoptotic pathways such as *Bid*, *Pten*, *Cdk2na,* and *Mgmt*, suggesting that inactivation of apoptosis drives tumor growth and metastasis [117].

Together, these screening systems have contributed enormously to our understanding of organismal and cell biology, as well as the biology of cancer cells. While some screening systems have lost popularity over newer and more sophisticated screening systems, each screen type comes with its advantages and disadvantages, as summarized in Table 1.

## 6. Conclusions and Future Perspectives

In this review we discussed how reverse genetic approaches and small-scale forward genetics screens in aneuploidy models have been used to determine how cells adapt to aneuploidy. Aneuploidy is mostly detrimental for cells and initially leads to a proliferative disadvantage, presumably due to the activation of aneuploidy-imposed stress pathways. It is therefore likely that aneuploid cells, throughout their malignant transformation process, need to overcome these stresses. Therefore, the molecular mechanisms underpinning these aneuploidy-induced stresses are considered to be promising therapeutic targets. The work of many labs in the last 15 years has significantly improved our understanding of some of the roadblocks that aneuploid cells need to overcome during tumorigenesis. However, to our knowledge, no large-scale screens have been reported that systematically compare the pathways affected in aneuploid cancers to the those affected in euploid cancers. When performed in an isogenic setting, such screens would surely reveal the differences between euploid and aneuploid cells on their route to a malignant program.

In this review, we discussed five types of mutagenesis screens that could be suitable for this goal, each with their own advantages and disadvantages (Table 1). ENU mutagenesis could be very effective in screening for point mutations that would accelerate the transformation of aneuploid cells. However, identifying the individual mutations that drive the phenotype is extremely laborious and many mice would be needed when such a screen would be performed in vivo. Retroviral mutagenesis allows for rapid identification of the mutated gene that improves the survival of aneuploid cells. However, these screens only sample proliferative tissues as the virus will only integrate in dividing cells. Because of this important limitation, retroviral tagging screens have mostly been surpassed by transposon, RNAi, and CRISPR screens. Indeed, transposon mutagenesis can be induced in any cell type within the whole organism, using a ubiquitously expressed transposase or in individual tissues with a conditional transposase controlled by a tissue-specific Cre-recombinase. Transposon mutagenesis furthermore allows for the identification of multiple collaborating driver mutations, which more accurately reflects the complexity of human cancer than a single mutation. However, transposons do display some insertion site preference, which yields to some bias in the screened part of the genome. This problem was largely overcome with the introduction of PiggyBac transposons, which suffer less from ‘local hopping’ and thus target the whole genome more efficiently [77]. In CRISPR/Cas9 and RNAi interference screens, such bias can be eliminated by careful sgRNA/shRNA/RNAi library design. RNAi have lost some popularity at the benefit of CRISPR screens, as CRISPR screens completely inactivate the targeted genes instead of (partially) knocking gene expression down and display fewer off-target effects. Moreover, CRISPR genome engineering offers many more applications, such as knockdown, knockout, knock-in, activation, and base editing [118,119], all of which can be exploited in genetic screens.

Altogether, to identify in an unbiased fashion the changes needed to convert an aneuploid cell into a cancer cell, one would need to setup an in vivo screen that would compare tumorigenesis in an euploid and aneuploid background. As stable aneuploidy is probably not sufficient to accelerate cancer in mice, the aneuploid background would need to be generated by crossing the ‘screening mice’ into a well-characterized CIN-predisposed background. This would likely work well as in many mouse models for CIN-driven cancer, CIN alone is not a powerful driver of cancer, but rather an accelerator [120,121]. Given that ongoing CIN is incompatible with early embryonic development [120], the most suitable CIN predisposition would be a conditional CIN-driving allele that does not efficiently promote cancer by itself. This could for instance be a *Mps1* truncation or mutation allele [33,122], a *Mad2* deletion allele [32], a hypomorphic BubR1 allele [123], or a Plk4 overexpression allele [24], as well as any other tissue-specific CIN driver. Indeed, in most of the CIN models, the CIN-driving allele alone leads to aneuploidy but not to rapid tumorigenesis. However, combining CIN with a single mutation in *p53* not only leads to cancer initiation [32,33,35,122,123,124] but also to a significant reduction of tumor latency, which makes this setup very suitable for a mutagenesis screen.

Altogether, we conclude that genome-wide mutagenesis screens in a CIN-predisposed background will likely yield important steps forward in the identification of more mechanisms of aneuploidy tolerance in vivo.

## Figures and Tables

**Figure 1 cells-10-00342-f001:**
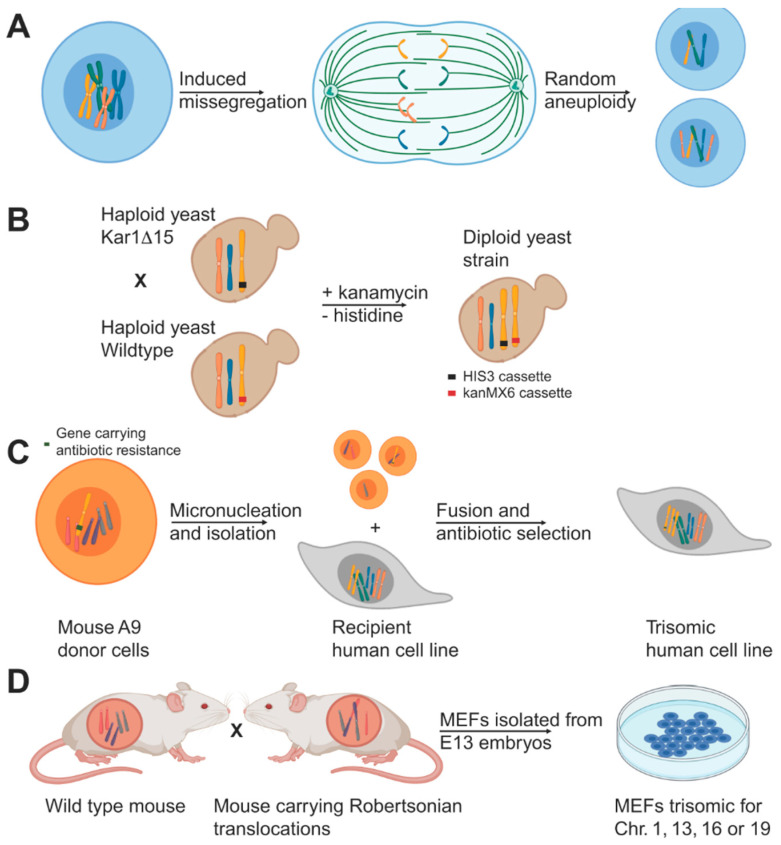
Examples of some of the aneuploidy model systems. (**A**) Induced chromosome mis-se-gregation by weaking the spindle assembly checkpoint using drug treatment or RNAi generates cells with random aneuploidies. (**B**) Disomic yeast cells can be generated by mating wild type yeast with yeast carrying a KAR1 mutation. Lack of KAR1 interferes with nuclear fusion and occasionally leads to random chromosome transfer and aneuploidy. Different antibiotic selection markers on the chromosomes allow for the selection of disomic cells in a haploid background. (**C**) Micronuclei-mediated chromosome transfer can be used to generate human aneuploid cells. For this, mouse donor cells carrying an additional human chromosome undergo micronucleation upon prolonged colcemid treatment and subsequent cell division. Isolated micronuclei carrying one or more chromosomes are fused with human acceptor cells. Aneuploid cells are selected for using a selection marker on the aneuploid chromosome. (**D**) Mating of wild type mice with mice carrying a Robertsonian translocation produces embryos with aneuploid karyotypes. Harvesting of cells from E12.5 embryos from these crosses allowed for the generation of trisomic MEFs.

**Table 1 cells-10-00342-t001:** The advantages and disadvantages of several mutagenesis systems.

Mutagenesis System	Advantages	Disadvantages
Chemical	Induces point mutations Unbiased disease gene discovery based on phenotyping Can be used in forward and reverse genetic approaches In vitro and in vivo use	Labor intensive positional cloning to identify mutated gene Identification of recessive genes in vivo requires back- or inter-crossing; many mice required Base pair substitution bias; some genes or domains more frequently mutated
Retrovirus	Rapid identification of mutated gene Does not require generation of transgenic mice for in vivo screens In vitro and in vivo use	Mostly identifies gain of function mutations (Most) cells must be dividing for retrovirus integration Strain-specific effects and limitations Limited tissue flexibility
Transposon	Genome-wide Loss and gain of function In vitro and in vivo use Allows for the identification of multiple cooperating mutations Can identify the effects of mutations in non-coding regions of the genome Can be done in vivo in whole organism or in tissue specific setup	Requires generation of transgenic lines Insertion site preference leading to bias SB has tendency for local hopping, and leaves footprint behind. Note that these disadvantages are not true for PB transposons Does not allow for identification of point mutations
RNA interference	Genome-wide Stable In vitro and in vivo use	Only loss of function Off target effects Does not identify multiple cooperating genetic mutations required for phenotype
CRISPR-Cas9	Genome-wide Can identify loss and gain of function mutations (CRISPRi/CRISPRa) In vitro and in vivo use Can be done in vivo in whole organism or in tissue specific setup	Does not identify multiple cooperating genetic mutations required for phenotype

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
