# Peer review of "Understanding How Genetic Mutations Collaborate with Genomic Instability in Cancer"

_cells, 2021, doi:10.3390/cells10020342_

Round 1
Reviewer 1 Report
In this review, Foijer and co-workers provide a very interesting compilation on recent progress and future challenges to uncover the “aneuploidy paradox”.
The topic is of broad interest and the review is overall very well written. I particularly appreciate the balance between conceptual advances along with experimental approaches/challenges. I am therefore favorable to its publication (even without the suggestions stated below).
I do, however, have a few comments that I would like the authors to consider:
- The authors summarize different models developed in the study of stable aneuploidies and provide the critical advantage of disentangling CIN from aneuploidy. Yet, it would be interesting to discuss the disadvantages of this approach. It is possible (likely) that the heterogeneity and karyotype diversity are intrinsically part of the aneuploidy tolerance behavior observed in cancer cells. This is somehow mentioned in Box 1, but considering the importance of this major caveat it would somehow be referred in the main text. Particularly as the screen section of the review is not very focused on stable aneuploidy models. Hence, a more coherent flow between the models presented and the ideal screen set-up would be advantageous.
- Box 1 appears with the text cut, so I may be missing some additional points. Yet, the text is somehow very centered on alleviating SAC. Several important studies have used other approaches (centrosome amplification, cohesion loss) and is unclear, given the broad scope of the review, why this section is exclusively focused on SAC alleviation.
- The review presents a selection of aneuploidy models (mostly focused on stable aneuploidies) that ranges from yeast to mammalian cell lines. This section, however, would be more complete if important advances made in other animal model systems would also been mentioned (e.g. Drosophila). Studies in fruit flies uncovered tissue-specific aneuploidy responses (e.g. premature differentiation (Gogendeau et al 2015; delayed onset of stress response (Mirkovic et al 2019); dysplasia (Resende et al 2018) or apoptosis (Poulton et al 2017)). Of note, one of these studies uses reversible mitotic perturbation (Mirkovic et al 2019) thereby uncoupling CIN from aneuploidy response, in line with what the authors state as a critical challenge in uncovering the true “aneuploid response”. This is just a suggestion as the authors choice of models to be presented and discussed should be respected. But in case they do not wish to provide a thorough description of all models available (which may indeed be a difficult taks) and focus on their own selection, the text should be modified to emphasize they are not summarizing ALL models available (e.g. title of figure 1 should state “examples of aneuploidy models”, etc).
- The “gene expression changes” should somehow include also some words on dosage compensation, as if present this process may alone contribute to direct changes in gene expression regulation, beyond the discussed changes related with stress response and/or metabolic alterations. This has remained a somehow controversial topic in the field yet if dosage compensation does exist (??) it may also be a potential “druggable” pathway to consider.
Author Response
Dear editor and reviewers
We would like to start by thanking the reviewers for their time and thoughtful review of our manuscript. We incorporated all of them and they improved the manuscript, in our opinion, a lot. Please find our detailed response below in blue.
Reviewer 1
In this review, Foijer and co-workers provide a very interesting compilation on recent progress and future challenges to uncover the “aneuploidy paradox”.
The topic is of broad interest and the review is overall very well written. I particularly appreciate the balance between conceptual advances along with experimental approaches/challenges. I am therefore favorable to its publication (even without the suggestions stated below).
Many thanks for your kind words.
I do, however, have a few comments that I would like the authors to consider:
1. The authors summarize different models developed in the study of stable aneuploidies and provide the critical advantage of disentangling CIN from aneuploidy. Yet, it would be interesting to discuss the disadvantages of this approach. It is possible (likely) that the heterogeneity and karyotype diversity are intrinsically part of the aneuploidy tolerance behavior observed in cancer cells. This is somehow mentioned in Box 1, but considering the importance of this major caveat it would somehow be referred in the main text. Particularly as the screen section of the review is not very focused on stable aneuploidy models. Hence, a more coherent flow between the models presented and the ideal screen set-up would be advantageous.
We see and appreciate the point the reviewer is making. While we agree that a more balanced description of models of stable aneuploidy vs models for ongoing CIN would be best, given the little time we were given for our rebuttal, we chose to better indicate that the emphasis of the model systems is on models for stable aneuploidy. However, to make the storyline more coherent, we incorporated the complete text of Box 1 into the main text and added some extra text to briefly discuss findings from CIN models. We believe that the overall text is now more coherent and describes both models for stable aneuploidy and ongoing CIN.
2. Box 1 appears with the text cut, so I may be missing some additional points. Yet, the text is somehow very centered on alleviating SAC. Several important studies have used other approaches (centrosome amplification, cohesion loss) and is unclear, given the broad scope of the review, why this section is exclusively focused on SAC alleviation.
Indeed, the reviewer is correct that the text was cut. The original text already referred to other drivers of CIN in models for ongoing CIN and are now also part of the main text as discussed under point 1.
3. The review presents a selection of aneuploidy models (mostly focused on stable aneuploidies) that ranges from yeast to mammalian cell lines. This section, however, would be more complete if important advances made in other animal model systems would also been mentioned (e.g. Drosophila). Studies in fruit flies uncovered tissue-specific aneuploidy responses (e.g. premature differentiation (Gogendeau et al 2015; delayed onset of stress response (Mirkovic et al 2019); dysplasia (Resende et al 2018) or apoptosis (Poulton et al 2017)). Of note, one of these studies uses reversible mitotic perturbation (Mirkovic et al 2019) thereby uncoupling CIN from aneuploidy response, in line with what the authors state as a critical challenge in uncovering the true “aneuploid response”. This is just a suggestion as the authors choice of models to be presented and discussed should be respected. But in case they do not wish to provide a thorough description of all models available (which may indeed be a difficult taks) and focus on their own selection, the text should be modified to emphasize they are not summarizing ALL models available (e.g. title of figure 1 should state “examples of aneuploidy models”, etc).
This is a fair point and indeed it is hard to include all findings from all model systems. We now included several references to the beautiful work done in Drosophila (apologies for our oversight) and also specifically referenced the ‘reversible CIN’ model in Drosophila. Furthermore, we made it much more explicit that we only discuss a selection of the model systems for CIN and aneuploidy.
4. The “gene expression changes” should somehow include also some words on dosage compensation, as if present this process may alone contribute to direct changes in gene expression regulation, beyond the discussed changes related with stress response and/or metabolic alterations. This has remained a somehow controversial topic in the field yet if dosage compensation does exist (??) it may also be a potential “druggable” pathway to consider.
Dose compensation is indeed a very interesting issue, which we now discuss briefly under ‘Gene expression changes’.
Reviewer 2 Report
In this paper, Jilderda et al. reviewed the available models to study aneuploidy in cancer cells, the principal aneuploidy-tolerating pathways and the genetic screens used for the identification of aneuploidy specific cancer driver genes.
The review is interesting and well structured but some criticisms should be addressed:
Box 1 is incomplete.
The authors should comment the opposite effect on nucleic acid metabolism and ribosomes obtained in aneuploid yest strains (ref 3) and aneuploid human cell lines (ref 13).
In the chapters dedicated to screening models, the authors should more focus on the identification of genes related to aneuploidy and aneuploidy-tolerating pathways rather than of general cancer genes.
In the Conclusions the authors stated that CRISPR screen is not expected to induce a strong immune response “in vivo” compared to RNAi. Nevertheless, references were not provided and the “in vivo” application of CRISPR was not enlisted in table 1.
English language should be revised. In particular several sentences are too long and convoluted.
Minor comments:
The conclusive sentences of each main chapter should be spaced in a separate paragraph.
The identical sentences were repeated in the Introduction and in the chapter entitled “Early finding from model system”.
In “Transposon mutagenesis” paragraph correct the sentence “Sanger OR Next Generation Sequencing”.
Table 1: The following sentence is in italics: “Base pair substitution bias; some genes or domains more frequently mutated”. In Transposon line, the two columns overlap.
Author Response
Dear editor and reviewers
We would like to start by thanking the reviewers for their time and thoughtful review of our manuscript. We incorporated all of them and they improved the manuscript, in our opinion, a lot. Please find our detailed response below in blue.
In this paper, Jilderda et al. reviewed the available models to study aneuploidy in cancer cells, the principal aneuploidy-tolerating pathways and the genetic screens used for the identification of aneuploidy specific cancer driver genes.
The review is interesting and well structured but some criticisms should be addressed:
Thank you for your kind words
Box 1 is incomplete.
Indeed, Box1 got cut when the manuscript was formatted by MDPI press. We now incorporated the text of Box1 in the main text, to get a more coherent story line.
The authors should comment the opposite effect on nucleic acid metabolism and ribosomes obtained in aneuploid yest strains (ref 3) and aneuploid human cell lines (ref 13).
This is a very interesting point that we previously missed. We now discuss this in the context of these two papers and some more papers, which suggest that yeast cells behave more as cancer cells.
In the chapters dedicated to screening models, the authors should more focus on the identification of genes related to aneuploidy and aneuploidy-tolerating pathways rather than of general cancer genes.
This is a fair point and we added a few more examples for several screen types. We did not find genome wide screens that asses aneuploidy tolerating genes, hence our conclusions that such screens need to be done.
In the Conclusions the authors stated that CRISPR screen is not expected to induce a strong immune response “in vivo” compared to RNAi. Nevertheless, references were not provided and the “in vivo” application of CRISPR was not enlisted in table 1.
This is a fair point too: we looked into this and while for shRNA and siRNA it has been reported that it triggers a strong interferon response, we also found some new reports that suggest that gRNAs might also trigger an inflammatory response. Since, to the best of our knowledge, these inflammatory responses have not been compared between siRNA/shRNA and gRNA, we decided to remove this statement from the text. We added the in vivo application of CRISPR as an advantage to Table 1.
English language should be revised. In particular several sentences are too long and convoluted.
We shorted several long sentences by breaking them up.
Minor comments:
The conclusive sentences of each main chapter should be spaced in a separate paragraph.
We made clearer conclusive paragraphs.
The identical sentences were repeated in the Introduction and in the chapter entitled “Early finding from model system”.
We changed the sentence in the introduction.
In “Transposon mutagenesis” paragraph correct the sentence “Sanger OR Next Generation Sequencing”.
We corrected this.
Table 1: The following sentence is in italics: “Base pair substitution bias; some genes or domains more frequently mutated”. In Transposon line, the two columns overlap.
We corrected these errors.
Round 2
Reviewer 1 Report
The authors have addressed most of my concerns and I am happy o recommend the publication of this interesting review.
Reviewer 2 Report
In this revised version of the paper the authors addressed most of the criticisms.
I have only minor comments:
The last sentence of the paragraph "Stable aneuploid human cell lines" requires a comma.
In the paragraph "Gene expression changes" the authors stated that multiple human tri-and tetrasomic cell lines were transcriptionally profiled and compared to aneuploid human cancer cell line. However, Dürrbaum et al. 2014 compared aneuploid cell lines with diploid progenitors. Moreover, the sentence "yeast cell biology represents the biology of an in vivo cancer cell more than the biology of cultured mammalian cells, at least in its response to aneuploidy" is too strong, I would rather say "at least for some responses to aneuploidy".
In the Conclusion correct the sentence: "Such screens WOULD, when performed in an isogenic setting WOULD surely reveal.."
In Table 1, the third disadvantage of Chemical is still in italics.